# Methotrexate Decreases the Level of PCSK9—A Novel Indicator of the Risk of Proatherogenic Lipid Profile in Psoriasis. The Preliminary Data

**DOI:** 10.3390/jcm9040910

**Published:** 2020-03-26

**Authors:** Julita Anna Krahel, Anna Baran, Tomasz W. Kamiński, Magdalena Maciaszek, Iwona Flisiak

**Affiliations:** 1Department of Dermatology and Venereology, Medical University of Bialystok, Zurawia 14 St., 15-540 Bialystok, Poland; aannabaran@wp.pl (A.B.); iflisiak@umb.edu.pl (I.F.); 2Department of Farmacodynamics, Medical University of Bialystok, Mickiewicza 2c St., 15-222 Bialystok, Poland; tomasz.kaminski@umb.edu.pl; 3Pittsburgh Heart, Lung and Blood Vascular Medicine Institute, University of Pittsburgh, Pittsburgh, PA 15260, USA; 4Department of Infectious Diseases and Hepatology Medical University of Bialystok, Zurawia 14 St., 15-540 Bialystok, Poland; mm.maciaszek@wp.pl

**Keywords:** psoriasis, proprotein convertase subtilisin/kexin type 9, methotrexate, acitretin, PCSK9, lipid metabolism

## Abstract

Background: Proprotein convertase subtilisin/kexin type 9 (PCSK9) exerts an important role in inflammatory processes, lipids homeostasis, and cardiometabolic disorders that are closely associated with psoriasis. The aim of the study was to analyze the clinical and diagnostic value of serum PCSK9 concentrations and their connections with disease severity, inflammation, metabolic syndrome, and impact of systemic therapies in psoriatic patients. The study enrolled thirty-five patients with active plaque-type psoriasis and eighteen healthy volunteers served as controls. Blood samples were obtained before and after 12 weeks of treatment with methotrexate or acitretin. Serum PCSK9 concentrations were measured by the ELISA (Enzyme-Linked Immunosorbent Assay) commercial kits. Morphological and biochemical parameters were assayed using routine laboratory techniques. Psoriatic patients showed significantly elevated levels of PCSK9 compared to controls (*p* < 0.01), mostly in patients with a mild and moderate course of psoriasis. PCSK9 concentrations correlated positively with BMI and triglyceride levels (*p* < 0.05). Interestingly, PCSK9 had a strong negative correlation with low-density lipoprotein levels and total cholesterol (*p* < 0.05). Three months of monotherapy with methotrexate significantly reduced PCSK9 level (*p* < 0.05), on the contrary, the acitretin group showed a further increase of PCSK9 levels (*p* < 0.05). PCSK9 seems to be a novel marker of psoriasis and a putative explanation of lipid disturbances, which are common in patients with psoriasis and are vital for the further developing of metabolic syndrome. Methotrexate should be considered as a treatment of choice in patients with an elevated PCSK9 concentration.

## 1. Introduction

Psoriasis is a systemic, immunometabolic, and still untreatable disease affecting 2%–4% of the general population worldwide. This disease was considered as a solely dermatological condition, while recent studies have linked psoriasis to many comorbidities namely obesity, cardiometabolic events, diabetes mellitus type 2 (DM type 2), lipid disturbances, and liver dysfunctions [1,2,3]. Further investigations have highlighted a higher risk of hypertension, metabolic syndrome (MS), or DM in association with the severity of psoriasis [1,2,3,4,5,6]. The morbidity of psoriatic patients is significantly higher compared with the population, which is due to the increased risk of cardiovascular events, mainly myocardial infarction (MI) and thromboembolic disorders [4,5,6,7]. However, the exact etiology of atherothrombotic events in psoriasis is still unknown. Common mechanisms which are taken into consideration are chronic inflammation, genetics, insulin resistance (IR), angiogenesis, oxidative stress, numerous bioactive molecules or proatherogenic lipids, and lipoprotein propensity [5,6,7,8]. The putative connection between CMDs and psoriasis have been also translated through psoriatic march, which means that psoriasis augments inflammation causing endothelial dysfunction and finally to atherosclerosis and further to cardiovascular events [9].

Lipid metabolism disturbances are vital cardiometabolic risk factors, closely related to psoriasis. Several reports have proven that psoriatic patients have proatherogenic lipoprotein propensities including hyperglyceridemia and a raised plasma concentration of low-density lipoprotein (LDL) and lowered high-density lipoprotein (HDL) concentrations [8]. Furthermore, autoantibodies targeting oxidized LDLs have been recognized in psoriasis and positive correlation with the disease severity has been reported [10]. So far, it is still debatable whether lipid disturbances are primary events or psoriasis-related events and the actual underlying mechanisms of dyslipidemia in psoriasis are still unclear [9,11,12]. Accumulating data suggest that pathological lipid profile and psoriasis have common pathogenic mechanisms. For instance, both diseases are strongly associated with T helper (Th) type 1-cell subset producing interferon-gamma (INF-gamma) or tumor necrosis factor alpha (TNF-alfa) [13]. Furthermore, Andreassen et al. hypothesized that systemic statins therapy decreases the risk of psoriasis through their pleiotropic interplay [14].

Finally, Seidah et al. described in 2003 an endogenous protein that directly affects LDL levels in the blood [15]. This discovered molecule proprotein convertase subtilisin/kexin type 9 (PCSK9), is a member of the family of serine proteinases named proprotein convertases (PCs). Members of the subtilisin/kexin proprotein convertase (PCSK) protease family transform inactive proproteins such as prohormones and cytokines into their biologically active forms [16]. PCSK9 was primarily referred to as “Neural apoptosis-regulated convertase-1” (NARC1), and was at first suspected to grants a congruous enzymatic activity like its other PC family members. A few years later it was discovered that PCSK9 has a special function of binding to the LDL receptor (LDLR) and causing its degradation [17,18]. The liver is the most important organ where PCSK9 is profusely expressed, but PCSK9 has been also discovered in numerous tissues such as lung, intestine, kidney, and brain [15,19]. Accurately, PCSK9 attaches to the LDLR on the cell surface and during the process of internalizations of the receptor, PCSK9 delivers the receptor complex to the endosome for demotion [20,21]. Finally, the whole operations causes a down-regulation of LDLR at the hepatocyte surface and leads to up growth of circulating levels of LDL and further to a higher risk of cardiometabolic diseases (CMDs). In the end, PCSK9 augments cardiovascular risk by reducing clearance of LDL and acting as a vital regulator of atherogenic inflammation due to attaching with toll-like receptors (TLRs) nuclear factor kappa-light-chain-enhancer of activated B cells (NF-kappaB) [22].

Recently there has been more research performed to comprehend the role of PCSK9 in other diseases besides dyslipidemia [23]. Current literature has pointed that expression and release of PCSK9 depends on other chronic conditions including chronic kidney disease, hyperinsulinemia, hypothyroidism or non-alcoholic fatty liver disease (NAFLD) and also inflammation, which is a significant factor in psoriatic pathogenesis [24,25]. As well as NAFLD which shares similar pathogenetic pathways, particularly metabolic or immunological, with psoriasis, with co-occurence in up to 50% of the patients [26,27], there is increasing interest to comprehend whether PCSK9 has a positive or negative impact on the inflammation in chronic and autoimmune diseases [28,29].

In 2018, in research conducted by Luan et al., the link between psoriasis and PCSK9 was proven. The authors found that PCSK9 is overexpressed in psoriatic-like lesions, provoked by imiquimod, in a mouse model [27].

Fang et al. recently suggested that elevated PCSK9 levels are probably associated with atherogenic inflammation in systemic lupus erythematosus (SLE) [13]. A putative link between psoriasis and PCSK9 activity could be found in research conducted by Cao et al. They demonstrated that elevated tumor necrosis factor alpha (TNF-alfa), one of the crucial pathogenetic factors in psoriasis, suppresses PCSK9 transcription in HepG2 (hepatocellular carcinoma cells) cells and in vivo [30]. On the other hand, oxidative stress is also taken into consideration in psoriasis pathogenesis. Shulter et al. described a positive correlation between oxidized low-density lipoprotein (oxLDL) and PCSK9 expression in the endothelium [31]. Elevated PCSK9 is also observed in metabolic syndrome, which affects more than 50% of psoriatic patients [24]. Furthermore, PCSK9 expression can be regulated by agonists adiponectin receptors (AdipoR agonists) through peroxisome proliferator–activated receptor gamma (PPAR-gamma) [32]. Reduction of circulating adiponectin levels and an expression of PPAR-gamma has been proven to be linked with obesity in psoriasis [32,33]. According to dual mechanisms in lipids metabolism and inflammation, PCSK9 monoclonal inhibitors termed ewolokumab and alirokumab had proven to be highly promising in bringing added cardiovascular benefit in the ODYSSEY LONG TERM trial and the FOURIER trial [34,35]. The clinical significance of alirocumab was assessed in high CV-risk patients and proved a valid reduction in plasma LDL (low density lipoprotein) and apoB levels compared to controls. Moreover, alirocumab showed significant LDL cholesterol (LDL-C) decreasing by up to 70% [36].

Other than the research cited above, the data linking PCSK9 to psoriasis are still limited. Our aim was to investigate PCSK9 levels in patients with active plaque-type psoriasis and its relation to the disease intensity, metabolic, or inflammation parameters and also to define an impact of systemic therapy. Furthermore, we aimed to measure the clinical significance of PCSK9 in psoriasis and its potential importance in estimating the probability of cardiometabolic events in patients with psoriasis with highlighted regard to proatherogenic lipid profile.

## 2. Materials and Methods

The study enrolled thirty-five patients (13 females and 22 males) with plaque-type psoriasis in a time of the exacerbation of the disease, at median age of 51 (19–78 years) and compare them with 18 sex-matched and age-matched healthy volunteers. All participants signed informed consents before initiation. None of the patients or controls was under any dietary restriction or was taking medications for at least three months before the enrollment. The exclusion criteria comprised of other types of psoriasis, chronic inflammatory diseases, and cardiometabolic, autoimmune, or oncological comorbidities.

Psoriasis area and its severity index (PASI) were assessed by the same person in all patients. The investigated group was divided depending on the disease intensity into three sub-groups: mild (PASI1) < 10 points, moderate (PASI2) between 10 and 20, and severe (PASI3) > 20. Body mass index (BMI) was calculated as weight/height^2^ (kg/m^2^). All subjects were further subdivided into groups according to BMI, BMI1 was related to normal-weight (BMI 18.5–24.9) and included 12 persons in group 2, overweight (BMI 25–29.9) was noted in 13 psoriatic patients; and BMI3, obesity (BMI > 30) was observed in 10 patients. Laboratory tests including C-reactive protein (CRP), complete blood count (CBC), serum glucose, total cholesterol (Chol), HDL, LDL, triacylglycerol (TAG), and transaminases were performed before treatment. The patients received two systemic treatment options: 15 persons took methotrexate (MTX) 15 mg/week using folic acid supplementation (15 mg/week, 24 h after MTX intake) and 20 subjects were started on acitretin at a dose 0.5 mg/kg/day. The treatment period lasted 12 weeks. The study was approved by the Bioethical Committee of Medical University in Bialystok (number: R-I-002/429/2017) and was in accordance with the principle of the Helsinki Declaration.

### 2.1. Serum Collection

Fasting blood samples were received from healthy subjects and patients before and after 12 weeks of treatment using vacutainer tubes with a clot activator. Samples were centrifuged at 2000× *g* for 10 min and preserved at −80 °C until analyses. PCSK9 levels were measured using an enzyme immunoassay kit supplied by Quantikine ^®^ (R&D Systems, Minneapolis, MN, USA). Optical density was read at a wavelength of 450 nm. The concentrations were measured by interpolation from calibration curves prepared with standard samples supplied by the manufacturer. All the tests were performed by the same person in standardized laboratory settings.

### 2.2. Statistical Analysis

The normally distributed data were presented as mean  ±  1SD (standard deviation), while the non-Gaussian data was presented as median (full-range). Normality of distribution was tested using a Shapiro–Wilk W test. The Student t-test or nonparametric Mann–Whitney test were used to compare differences between the psoriasis group and the control group. The χ2 test was used for categorical variables. The correlations were analyzed using Spearman’s Rank correlation analysis or Quasi-Newton and Rosenbrock’s regression analysis. Multiple regression analysis was performed based on previous results from Spearman’s Rank correlation analysis or Quasi-Newton and Rosenbrock’s regression analysis. A two-tailed *p*  < 0.05 was considered statistically significant. Computations were performed using GraphPad 6 Prism (GraphPad Software; La Jolla, CA, USA).

## 3. Results

The characteristics of patients and controls are listed in Table 1 and Table 2.

A total of 35 patients with active plaque-type psoriasis, 13 women and 22 men with the mean age of 51 (19–78 years) and 18 age- sex-matched healthy subjects were enrolled into the study. Differences concerning BMI in controls and psoriatics might be explained through the fact that obesity is common in psoriasis, especially in the moderate-to-severe course of the disease and thus leads to a further increase of cardiometabolic risk (Table 1 and Table 2).

The median value of BMI was 27.7 kg/m^2^ (17.6.9–44.4) (Table 1). The median of basal PASI scores was 17 (8.4–33.5) points and 10.2 (4.8–23.9.4) after treatment. In the study group 9 patients had mild psoriasis (PASI < 10), 14 had moderate (PASI 10–20), and 12 were diagnosed with a severe form (PASI > 20).

The median PCSK9 level in patients before treatment was 1.71 (0.73–2.32) ng/mL which was significantly higher (*p* < 0.01) than that in the controls: −1.45 (0.79–1.99) ng/mL (Figure 1).

PCSK9 did not correlate with psoriasis activity expressed by the PASI score in the patient’s group before treatment (Table 3).

After dividing the study group regarding PASI, the protein concentrations were significantly increased in all three sub-groups before treatment compared to the controls: PASI1 (*p* < 0.01), PASI2 and PASI3 (both *p* < 0.05) (Figure 2).

PCSK9 did not correlate with total BMI in patients before treatment (*p* > 0.05) (Table 3). PCSK9 concentration was the highest in obese psoriatic patients and the lowest in overweight ones, but without statistical significance (Figure 3).

Regarding lipid metabolism indicators, PCSK9 correlated negatively with total cholesterol and LDL with statistical significance (*p* = 0.048, *p* = 0.048, respectively) and also negatively with HDL, but without statistical significance in patients before treatment. PCSK9 correlated positively with TGs (triglycerides), without statistical significance (Table 3). There were no significant correlations with CRP, RBC (red blood cells), WBC (white blood cells), glucose level or liver enzymes activity before and after treatment (Table 3). Strong negative correlation with LDL increased after treatment (*p* = 0.0062), but decreased with total cholesterol (Table 3).

After twelve weeks of the systemic therapy the skin lesions in all studied patients improved. The median of total PASI score decreased from basal PASI 17 (8.4–33.5) to 10.2 (4.8–23.9) after total therapy (Table 2). The median PCSK9 significantly decreased after treatment (*p* < 0.01), remaining statistically higher than those in the controls (*p* < 0.01) with the value of 1.67 (0.72–2.24) (Figure 1). After division into sub-groups of patients treated with both drugs separately, serum PCSK9 concentration decreased after methotrexate (*p* < 0.05) (Figure 4).

Interestingly, therapy with acitretin showed a further significant increase of PCSK9 levels (*p* < 0.05) (Figure 4). Accordingly, to the severity of psoriasis after treatment, PCSK9 remained significantly increased in all PASI sub-groups versus the controls (Figure 5). With regard to BMI, PCSK9 levels were higher in all three sub-groups compared to the controls and the significance was still the highest in the BMI1 and BMI3 sub-groups (Figure 3). PCSK9 levels significantly decreased after treatment in the obese psoriatic patients (Figure 3).

## 4. Discussion

Psoriasis is a chronic, T-cell-mediated inflammatory disease associated with numerous metabolic disorders, including dyslipidemia. PCSK9 is mainly known as a molecule negatively affecting lipid metabolism, leading to hypercholesterolemia and consequent atherosclerotic plaque formation. Moreover, PCSK9 exerts a pleiotropic effect due to stimulation and sustenance of chronic inflammation as well as prooxidative status redox enhancing an imbalance in homeostasis [37].

Based on the available literature’s data, our study is the first one assessing serum PCSK9 concentrations in patients with psoriasis and evaluating the impact of conservative and well-established treatments. A number of studies addressed the potential role of PCSK9 in systemic metabolism, obesity, and other CMDs [24,25]. To our knowledge, there is only one paper investigating PCSK9 in psoriasis. Luan et al. proved overexpression of PCSK9 in psoriatic lesions, indicating its important role in psoriasis, however the distinct relationship with lipid disorders should be further explored [27]. After considering these data, we maintained the thesis that PCSK9 seems to play a role in psoriasis and its association with the proatherogenic lipid profile, however further investigations are needed.

In the present study, we found that serum PCSK9 concentration was significantly elevated in psoriatic patients compared to the controls. Luan et al. suggested a possible role for PCSK9 in the course of this disease and further, proved that suppression of PCSK9 inhibits hyperproliferation of keratinocytes through inducing apoptosis [27]. According to these and our own results, we assume that PCSK9 is a novel biomarker of psoriasis, but not of its severity. There is no more related data concerning psoriasis and PCSK9. Due to the growing interest of the potential role of PCSK9 in many fields of medicine, there are a number of studies showing a crucial role of this molecule in other systemic diseases, but the data shows high levels of divergence. For instance, a meta-analysis performed in 2016 reported that PCSK9 levels were remarkably associated with an increased risk of cardiovascular events [38]. Although, another meta-analysis published one year later, found no associations between the same parameters [39]. Fang et al. demonstrated significantly elevated PCSK9 levels in systemic lupus erythematosus, especially with coexisting lupus nephritis (LN) and presumed its role in atherogenic inflammation in SLE. They found a positive correlation between PCSK9 and CRP, but not with lipid parameters or BMI, suggesting that PCSK9 impacts inflammation in SLE, but independent of atherogenic lipids [13]. The lipids exert deplorable effects towards endothelial cells, effectiveness nitric oxide activity, the development of a procoagulant surface, chronic low-grade inflammation, and abnormal cell growth [13]. Notably, in our research, there was no correlation between PCSK9 and CRP. This assumption sheds new light into the real diagnostic and therapeutic significance of the protein, which is not a reliable marker of the inflammatory state in psoriasis. This is in line with the results of a meta-analysis conducted by Cao et al. They demonstrated that inhibition of PCSK9 had no significant impact on high-sensitivity CRP (hs-CRP) [40]. On the other hand, Cui et al. proved direct interplay between CRP and PCSK9, thus CRP increases PCSK9 expression by activating p38 mitogen activated protein kinase (p38MAPK)-hepatocyte nuclear factor 1alfa (HNF1alfa) pathway in HepG2 cells [41]. Moreover, Rannikko et al. demonstrated that PCSK9 is upregulated in blood culture-positive infections and correlates positively with CRP levels, resembling acute-phase proteins [42]. Our results might indicate that increased PCSK9 levels are not directly involved in chronic inflammation in psoriatic patients, especially in the BMI1 and BMI2 sub-groups and there must be other interfering relationships. What is also worth mentioning here are the interesting results of the links between PCSK9 levels and inflammatory markers in rheumatoid arthritis (RA). A recent clinical study revealed that RA patients exhibited lower PCSK9 serum levels than healthy controls, which became contradictory to previous findings [43]. This confirms that lower PCSK9 levels do not exert anti-inflammatory effects in the RA model. Moreover, it proves that PCSK9 attributions seem to undergo further modifications by currently unknown factors [43].

Furthermore, Haas et al. showed increased PCSK9 levels in patients with chronic kidney disease and nephrotic syndrome, diseases strongly associated with a proinflammatory environment [44]. Furthermore, NAFLD, closely related to psoriasis, increases circulating PCSK9 concentrations independently of metabolic modifiers [42]. Interestingly, PCSK9 levels are decreased in patients with type 1 as compared to type 2 diabetes, whereas it has been proven that hyperinsulinemia increases PCSK9 expression in murine models [45,46,47].

In the present study, PCSK9 correlated negatively with total cholesterol and LDL, however, it correlated positively with TAG. It suggests that elevated PCSK9 levels are involved in lipid metabolism in psoriasis but are not directly linked to variations in LDL levels. LDLR-mediated PCSK9 can be degraded in both an intracellular and an extracellular way, and the exact LDL level depends solely on the PCSK9 component secreted in the extracellular pathway [48]. This phenomenon can be explained by the previous paper, stressing that even in the general population there are variations in PCSK9 concentrations [49]. Thus, on the one hand it is well documented that PCSK9 plays an important role in the elevation of plasma LDL levels that are related to the development of CVD. On the other hand, there are several recent reports that are consistent with the results of our study. Fang et al. did not find a correlation between PCSK9 levels and LDL, HDL, nor TAG [13]. Further, Liu et al. demonstrated a positive correlation between plasma PCSK9 and LDL in the general population’s cohort, but not in the type 2 diabetic subpopulation [50,51]. Similar results were reported by Zhang et al. that plasma LDL was only positively related to PCSK9 in patients with stable coronary artery disease [52]. It suggests, again, that the exact role of PCSK9 is still unclear, and that predictions of its action must be subject to many factors.

We found no correlations between PCSK9 and markers of liver function. This might reflect rather an uncertain role of PCSK9 in predicting liver damage in psoriatic patients. Our results are in line with the study conducted by Wargny et al. who reported no significant associations between PCSK9 and transaminases in hepatic steatosis and NASH in high-risk patients with coexisting DM type 2 and obesity [52]. However, Zaid et al. demonstrated the protecting and even regenerating role of PCSK9 in transgenic mice, suggesting that upon hepatic damage, patients lacking PCSK9 could be at risk [53].

Regarding demographic and clinical or demographical measurements, our results did not reveal any associations between PCSK9 and disease duration nor age, height, or gender of the study group. Interestingly, it correlated negatively with BMI after the treatment. Fang et al. demonstrated a positive correlation between PCSK9 levels and age in SLE patients [13]. In the recent study conducted by Jeenduang et al., PCSK9 concentrations were significantly higher in women than in men, especially in postmenopausal women [54]. Similar data were also reported by Cui et al. and Lakoski et al. [41,48]. In contrast, the Canadian study group did not show such associations [55]. Variations in PSCK9 among various studies may be due to differences in ethnicity as well as pathophysiological or hormonal conditions.

To our knowledge, there are no published research papers regarding systemic antipsoriatic treatment and its presumptive impact on PCSK9 levels. Both methotrexate and acitretin have been widely used against moderate-to-severe psoriasis with significant clinical results for many years. We investigated the effects on PCSK9 levels by monotherapy with methotrexate or acitretin. Previous studies have demonstrated significant reductions in CVD-related mortality in patients treated with MTX [56]. This finding has been linked to the anti-inflammatory properties of the drug because it reduces white blood cell growth and accelerates their apoptosis. Moreover, MTX decreases interleukin-1 and -6 secretion and at the same time increases interleukins-4 and -10 gene expression and decreases gene expression of proinflammatory Th1 cytokines [57]. In research conducted by Dehpouri et al., the scientists highlighted that MTX does not negatively affect the glucose metabolism in patients with psoriasis arthritis and can be safely used in the treatment of psoriasis and psoriatic arthritis, especially with coexisting diabetes mellitus and metabolic syndrome [58]. Acitretin possess a unique role in the treatment of psoriasis because its mechanism of action is different from other systemic drugs [59]. Cytosolic retinoic acid-binding proteins, which transport retinoic acid to the cell nucleus, is markedly elevated in psoriatic lesions as compared to being unaffected in skin lesions, indicating a greater sensitivity of the skin to the retinoid. Adverse effects of acitretin include dose-related changes in the lipid metabolism. A number of studies have shown an increase in triglycerides and cholesterol in 66% and 33% of patients, respectively, and a decrease in high-density lipoprotein in up to 40% of patients during acitretin therapy [60].

After 12 weeks of systemic therapy in total the median PCSK9 significantly decreased remaining statistically higher compared to the controls, as well as in each of the PASI sub-groups. This outcome shows that despite the potential implication of PCSK9 in the modulation of the risk of CMDs, the interplay with the therapy is still insufficient. We proved that PCSK9 levels cannot be used as a marker of effectiveness of psoriasis treatment. Interestingly, after division into sub-groups of patients treated with both drugs separately, serum PCSK9 concentration decreased after methotrexate, although, therapy with acitretin showed a further increase of PCSK9 levels. According to pleiotropic activity of PCSK9, we can speculate that in psoriatic patients with an initially increased level of PCSK9, MTX should be taken into account as a first-line treatment. Furthermore, MTX should be considered as a drug modulating pathways dependent on PCSK9, however further studies should be performed.

BMI levels and systemic therapy exert high influences on associations between PCSK9 concentrations and levels of HDL, LDL, CRP and ALT (alanine transferase). In the BMI1 sub-group, systemic treatment switched strong negative correlation with HDL into mild, positive correlation. It could be interpreted as another pleiotropic effect of PCSK9 in this particular sub-group. Further, the LDL level in the BMI2 sub-group had a significantly negative correlation with PCSK9 concentration before and after treatment. On the other hand, in the BMI3 sub-group, systemic treatment revealed strong negative correlation with LDL, which indicates that systemic treatment in obese psoriatic patients could exert more favorable effects. Presumably, connecting antipsoriatic treatment additionally with other lowering PCSK9 medications could be useful, especially when acitretin is used as a systemic drug in obese psoriatic patients with metabolic syndrome. Worth mentioning are recent studies on PCSK9-inhibition. As a result of rigorous research in this direction, fully human PCSK9-binding antibodies evolocumab and alirocumab were discovered and approved by the FDA (US Food and Drug Administration) in 2015. These studies reported an additional reduction of LDL by 50%–60% and a reduction of lipoprotein(a) (LPA) by 25%–30% [25]. Furthermore, in the ODDYSSEY LONGTERM study, randomized 2341 patients at high risk of cardiovascular events who received statins to alirocumab to placebo demonstrated that LDL was reduced by 62% at week 24 [35]. In another trial, OSLER, evolocumab reduced LDL by 61% [60].

The fundamental limitation of our study is the relatively small quantity of patients and their allotment in three independent subgroups which enhances the risk for bias occurrence and significantly reduced the statistical reliability of performed analyses. The present study has some other putative disadvantages that should be pointed out. We are aware that our results have to be pondered as a preliminary report due to the amount of analysed parameters and many consolidated factors are the main weak points in the conducted study. The analysed protein PCSK9 has multidirectional activity towards many organs and significantly augments theirs functions, which could entail a revaluation of its factual function in psoriasis. At this moment, the results of the preliminary study have to be considered as possible signposts leading to a better comprehension of the discussed issues.

## 5. Conclusions

To conclude, our study for the first time showed that PCSK9 levels are significantly increased in psoriatic patients and seems to be a novel marker of psoriasis and risk assessment of cardiometabolic disorders in this dermatosis. Further, we speculate that measuring PCSK9 levels could be useful in choosing the best psoriasis therapy method. Methotrexate should be the first-line treatment in patients with elevated PCSK9 concentration.

## Figures and Tables

**Figure 1 jcm-09-00910-f001:**
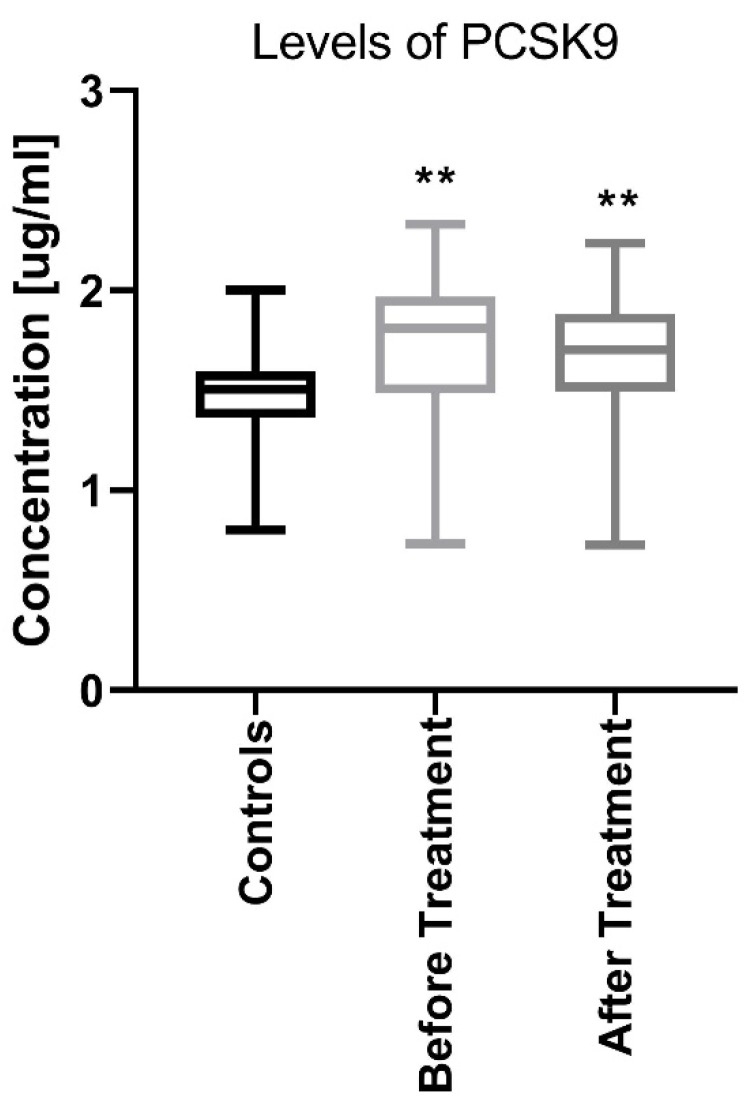
Comparison between PCSK9 concentrations before and after treatment in psoriatic patients and controls. ** means statistical significance with *p* < 0.01 compared to Controls. PCSK9, proprotein convertase subtilisin/kexin type 9.

**Figure 2 jcm-09-00910-f002:**
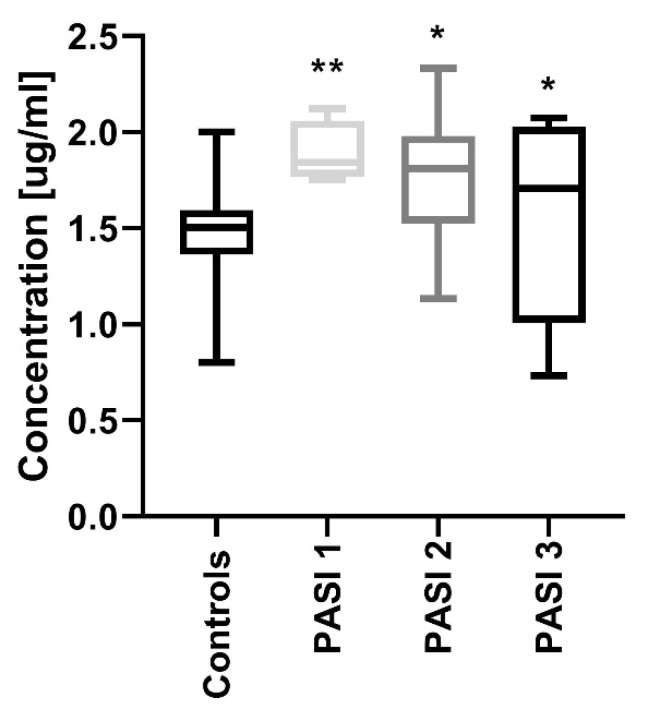
Comparison between PCSK9 concentrations depending on the psoriasis area and the severity index (PASI) before treatment and controls. */** means statistical significance with *p* < 0.05/0.01 respectively, compared to Controls.

**Figure 3 jcm-09-00910-f003:**
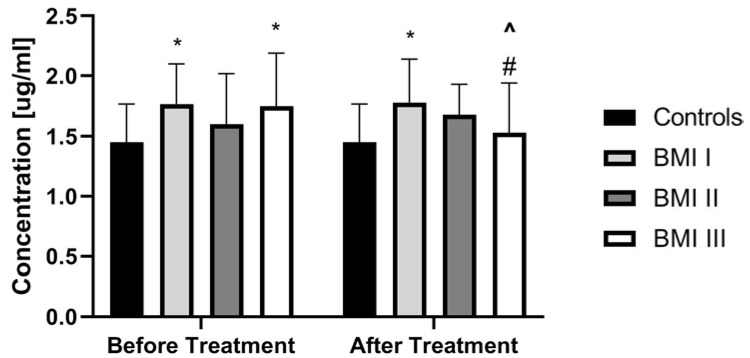
Comparison of PCSK9 concentrations between controls and patients sub-groups divided on a base of BMI scoring before and after treatment. * means statistical significance with *p* < 0.05 compared to Controls. # means significance with *p* < 0.05 compared to BMI 1 and BMI 2 subgroups. ^ means statistical significance with *p* < 0.05 for BMI 3 before treatment vs the same group after treatment. BMI, body mass index.

**Figure 4 jcm-09-00910-f004:**
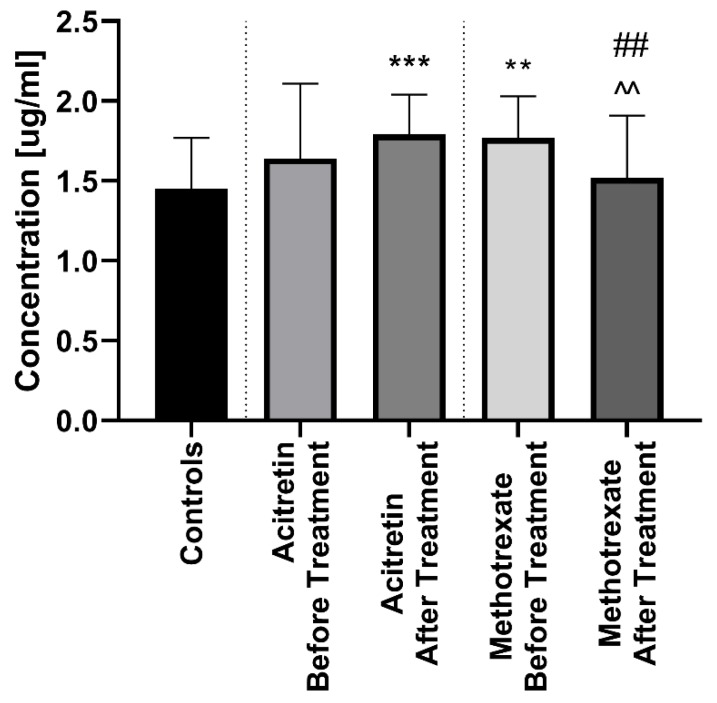
Comparison between PCSK9 concentations before and after treatment with acitretin and methotrexate separetely in psoriatic patients and controls. **/*** means statistical significance with *p* < 0.01/0.0001 respectively compared to Controls. ## means statistical significance with *p* < 0.01 between acitretin after treatment compared to methotrexate after treatment. ^^ means statistical significance with *p* < 0.01 between methotrexate before treatment compared to methotrexate after treatment.

**Figure 5 jcm-09-00910-f005:**
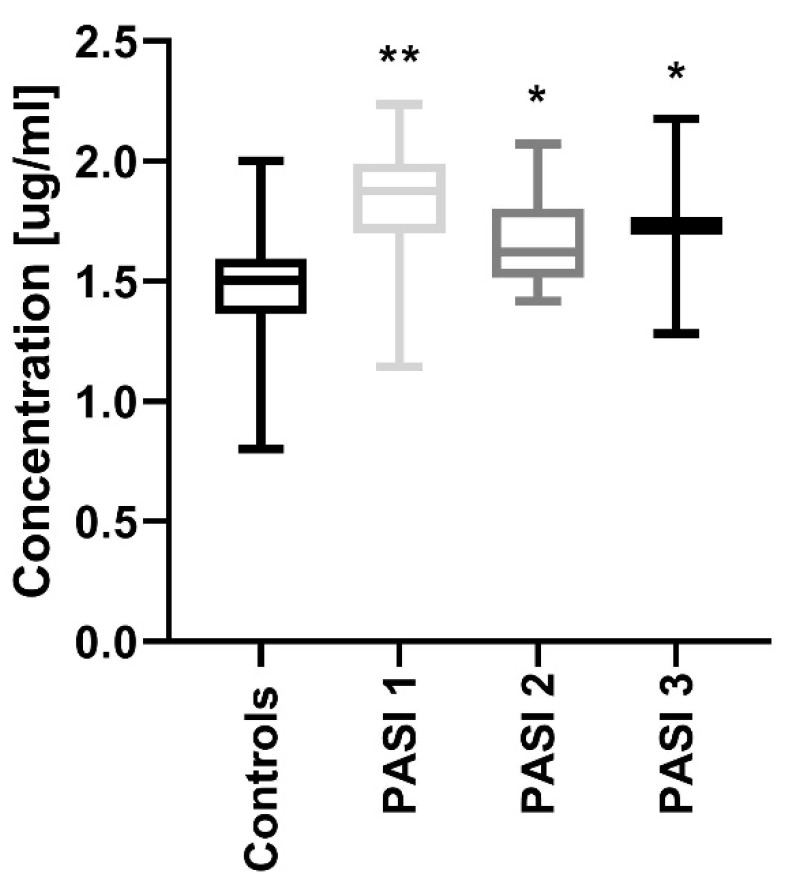
Comparison between PCSK9 concentrations depending on psoriasis area and severity index (PASI) after treatment and controls. */** means statistical significance with *p* < 0.05/0.01 respectively, compared to Controls.

**Table 1 jcm-09-00910-t001:** Basal characteristic of control group and patients group.

Parameter	Controls (*n* = 18)	Patients (*n* = 35)
Sex (M/F)	9/9	22/13 NS
Age (years)	30.5 (25–64)	51 (19–78) NS
Height (cm)	173 ± 9.5	174 ± 9.2 NS
Weight (kg)	72.5 ± 14.4	83.6 ± 16.2 *
BMI ratio	23.4 (20.1–32.7)	27.7 (17.6–44.4) **

*/** - statistical significance with *p* values <0.05 and <0.01, respectively, between Controls and Patients groups. NS, non-significant; M/F, male/female ratio; BMI, body mass index.

**Table 2 jcm-09-00910-t002:** Basal characteristic of patients group undergoing treatment.

Characteristics	Values Before	Values After
PASI score	17 (8.4–33.5)	10.2 (4.8–23.9) ***
RBC (×10^3^/mL)	4.61 ± 0.56 ^	4.67 ± 0.54 ^
PLT (×10^3^/mL)	230 ± 65.5 ^	232 ± 57.9 ^
WBC (×10^3^/mL)	7.15 (4.11–12.7)	6.42 (4.4–12.5)
Glucose level (mg/dL)	83 (66–229)	88 (69–250)
Cholesterol Total (mg/dL)	168 ± 28.7 ^	180 ± 32.1 ^
TGs (mmoL/L)	123 ± 49.6 ^	149 ± 71.8 (0.082) ^
HDL (mmoL/L)	47 ± 11.9 ^	47.7 ± 19.3 ^
LDL (mmoL/L)	103 ± 23.8 ^	106 ± 26.5 ^
CRP (mg/L)	2.95 (1–58.7)	1.9 (1–15.2) *
ALT (U/L)	17 (8–78)	18 (7–113)
ASPAT (U/L)	19 (12–71)	18 (11–114)

*- statistical significance with p values <0.05 after treatment compared to the values. *** - statistical significance with *p* values <0.001 after treatment compared to the values. ^ mean  ±  1SD (standard deviation), non-Gaussian data as median (full-range) in brackets. PASI, psoriasis area and severity index; RBC, red blood cells; PLT, platelets; WBC, white blood cells; TGs, triglycerides; HDL, high-density lipoproteins; LDL, low-density lipoproteins; CRP, c-reactive protein; ALT, alanine transaminase; ASPAT, asparagine transaminase.

**Table 3 jcm-09-00910-t003:** Variables of the study in patients before and after treatment and correlations with PCSK9 levels. Bold values mean statistically significant results.

Parameter / PCSK9	Before	After
Sex (M/F)	0.291 (NS) −0.052 (NS) 0.241 (NS) 0.127 (NS) 0.183 (NS)	
Age (years)		
Weight		
Height		
BMI		
PASI score	−0.15 (NS)	−0.271 (NS)
RBC (×10^3^/mL)	0.078 (NS)	−0.153 (NS)
PLT (×10^3^/mL)	0.025 (NS)	−0.111 (NS)
WBC (×10^3^/mL)	0.207 (NS)	−0.093 (NS)
Glucose level (mg/dL)	0.278 (NS)	−0.208 (NS)
Cholesterol Total (mg/dL)	−0.348 (0.048)	−0.185 (NS)
TGs (mmoL/L)	0.045 (NS)	0.062 (NS)
HDL (mmoL/L)	−0.267 (NS)	0.125 (NS)
LDL (mmoL/L)	−0.352 (0.048)	−0.535 (0.0062)
CRP (mg/L)	0.132 (NS)	−0.072 (NS)
ALT (U/L)	−0.016 (NS)	−0.108 (NS)
ASPAT (U/L)	−0.092 (NS)	−0.067 (NS)

PCSK9, proprotein convertase subtilisin/kexin type 9.

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
