# Peer review of "Methotrexate Decreases the Level of PCSK9—A Novel Indicator of the Risk of Proatherogenic Lipid Profile in Psoriasis. The Preliminary Data"

_jcm, 2020, doi:10.3390/jcm9040910_

Round 1

Reviewer 1 Report

The authors present an interesting and original paper, with an overall well-documented methodology. There are a few comments/suggestions to offer:

  1. Abstract, line 24: please explain PASI1 and PASI2 subgroups (or rephrase the terms, for example with "mild" and "moderate" disease severity) in order to achieve clarity for the physician that reads the abstract
  2. Abstract, line 28: "PCSK9 seems to be a novel marker of psoriasis and risk assessment of metabolic disorders in psoriatic." Although the data provided demonstrates a correlation between PCSK9 and CHOL, LDL, TAG, this should not be translated into interpreting PCSK9 as a marker of risk assessment of metabolic disorders in general in psoriasis patients. Please consider rephrasing.
  3. Introduction, line 104: "despite the existence of a strong background, there is lack of evidence linking 104 PCSK9 to psoriasis". Data exist linking psoriatic-like lesions to PCSK9 (Luan et al 2018), so please consider rephrasing, for example "there is (very) limited data..."
  4. Materials and Methods, line 142: please specify the meaning of "CKD" group, do you mean psoriasis patients?
  5. Results, Table 1A: In line 112 you report BMI-matched groups (psoriatic patient and control group) but there is a statistically significant difference presented in table 1.A. Please specify and correct accordingly.
  6. Results, Table 1B: please note when you use range or SD next to each characteristic. Also include explanation of abbreviations
  7. Results, line 159: "Differences concerning BMI in controls and psoriatic are due to the increased cardiometabolic risk in psoriasis (Table 1A and 1B)". Differences in BMI between the control group and the psoriasis patients are due to the increased BMI, which is common in psoriasis patients with moderate-to-severe disease. Please re-evaluate the meaning of this sentence.
  8. Results: the results presented in this paper are interesting. Did you explore the presence of psoriatic arthritis and how this comorbidity could influence the results?
  9. Discussion, line 232: "After considering these data, we maintained the thesis that PCSK9 seems to be involved in pathogenesis of psoriasis itself and also could have respectful prognostic value for mortality and morbidity in psoriasis due to CMDs". To this point, there is not enough evidence to state that PCSK9 is involved in the pathogenesis of psoriasis. Please consider rephrasing.
  10. Discussion, line 239: "Moreover, our outcomes are among the first ones, which directly connect levels of PCSK9 to the risk of the occurrence of comorbidities in psoriatic patients. There is no more related data concerning psoriasis...". Please specify which comorbidities you are referring to. This manuscript demonstrated the presence of elevated PCSK9 in patients with psoriasis, independently from disease severity (mild, moderate, severe) but no risk of developing any known comorbidities of psoriasis (hazard ratio analysis etc). The correlation to CHOL, LDL, HDL should be carefully interpreted due to the limited number of patients.
  11. Discussion, line 253: "there was no correlation between PCSK9 and CRP, suggesting that elevated concentrations of PCSK9 in psoriatics are not linked directly to the severity of the disease.". It is well known that CRP is also not always directly linked to the severity of skin psoriasis. Patients with moderate-to-severe skin psoriasis could also have normal CRP levels. Therefore, a lack of correlation between PCSK9 and CRP does not equal a lack of correlation between PCSK9 and disease severity. The lack of correlation between PCSK9 and CRP was demonstrated in this manuscript independently. Please proof the meaning of this sentence.
  12. Discussion: this section was in some paragraphs difficult to follow (for instance paragraphs 5 and 6), please consider rewriting in order to achieve more clarity.
  13. Please carefully proof-read the manuscript to eliminate grammatical errors and minor typos (for instance line 29 (abstract), 87 (introduction: "imiquimod" instead of "imikwimod"), 290, 299, 316 (discussion) etc.)

Author Response

Answer to the Reviewer 1.

Thank you very much for your valuable time and reviewing the manuscript. The manuscript has been supplemented with detailed suggestions. All changes are matched with red color.

  1. According to your suggestion we have changed “PASI” to more clinical nomenclature as moderate and mild course of psoriasis.
  2. The whole sentence have been rephrased accordingly to your proposition, and also I have proposed the new title of the manuscript.
  3. The sentence was rephrased.
  4. We specified – psoriatic patients.
  5. We removed BMI-matched from the manuscript.
  6. The table has been modified according to your suggestion.
  7. Changed sentence accordingly to the suggestion.
  8. Coexisting, diagnosed by rheumatologist, psoriatic arthritis was the exclusion criterium. We focused solely on plaque psoriasis. However in further research we’d like to widen our study group onto other types of psoriasis, including arthritic one.
  9. The sentence have been rephrased.
  10. Comorbidities were specified only to coexisting lipids disturbances.
  11. The sentence has been removed from the paper.
  12. We rewrote the order to achieve more clarity.

Reviewer 2 Report

It is a really interesting preliminary study toward the effect on MTX in PCSK9, a marker of CVD but also to great interest in these days since a targeting drug was recently introduced on the market.

Introduction
Line 38: Please add in the references (10.3390/jcm9010186, 10.1155/2018/3140983)
Please cite directly alirocumab and its mechanism of action

Material and Methods
Please define active psoriasis: patients were stable?
Please add a paragraph stating that the study was approved by your ethical committee and all patients signed a consent form.

Results: Coherent and well explained

Discussion
Please discuss the liver effect of methotrexate and how it may modulate also the glycemic effect (10.4081/dr.2019.7965)

Please add the author contribution clearly, funding sources and conflict of interest

Author Response

Answer to the Reviewer 2.

Thank you very much for your valuable time and reviewing the manuscript. It has been supplemented with detailed suggestions. All changes are matched with blue color and the’ve been put in the manuscript.

Once again, thank you for your contribution to our research. Kind regards. Julita Anna Krahel, MDCorresponding author,Dermatology and Venereology DepartmentMedical University of Bialystok, Poland